# Relationship between Treatment Plan Dosimetry, Toxicity, and Survival following Intensity-Modulated Radiotherapy, with or without Chemotherapy, for Stage III Inoperable Non-Small Cell Lung Cancer

**DOI:** 10.3390/cancers13235923

**Published:** 2021-11-25

**Authors:** Isabel F. Remmerts de Vries, Merle I. Ronden, Idris Bahce, Femke O. B. Spoelstra, Patricia F. De Haan, Cornelis J. A. Haasbeek, Birgit I. Lissenberg-Witte, Ben J. Slotman, Max Dahele, Wilko F. A. R. Verbakel

**Affiliations:** 1Department of Radiation Oncology, Amsterdam University Medical Centers, Vrije Universiteit Amsterdam, Cancer Center Amsterdam, 1081 HV Amsterdam, The Netherlands; m.ronden@amsterdamumc.nl (M.I.R.); i.bahce@amsterdamumc.nl (I.B.); f.spoelstra@amsterdamumc.nl (F.O.B.S.); pf.dehaan@amsterdamumc.nl (P.F.D.H.); cja.haasbeek@amsterdamumc.nl (C.J.A.H.); bj.slotman@amsterdamumc.nl (B.J.S.); m.dahele@amsterdamumc.nl (M.D.); w.verbakel@amsterdamumc.nl (W.F.A.R.V.); 2Department of Epidemiology and Data Science, Amsterdam University Medical Centers, Vrije Universiteit Amsterdam, Cancer Center Amsterdam, 1081 HV Amsterdam, The Netherlands; b.lissenberg@amsterdamumc.nl

**Keywords:** NSCLC, dosimetric parameters, overall survival, toxicity

## Abstract

**Simple Summary:**

Various radiotherapy treatment methods are available for patients with stage III non-small-cell lung cancer (NSCLC). A multidisciplinary tumor board review is recommended to determine the best treatment strategy. In fit patients with inoperable tumors, concurrent chemoradiotherapy (cCRT) is preferred over sequential CRT (sCRT), due to better survival. Nonetheless, the use of cCRT in stage III NSCLC varies significantly, with concerns about treatment toxicity being a contributory factor. Many reports describing the relationship between overall survival, toxicity, and dosimetry in patients with locally advanced NSCLC are based on clinical trials, with strict criteria for patient selection, including good performance status, pulmonary function, etc. These trials have not always mandated the use of IMRT/VMAT. We therefore performed an institutional analysis to study the relationship between dosimetric parameters and overall survival and toxicity in patients with stage III NSCLC treated with IMRT/VMAT-based techniques in routine clinical practice.

**Abstract:**

Concurrent chemoradiotherapy (cCRT) is the preferred treatment for stage III NSCLC because surgery containing multimodality treatment is often not appropriate. Alternatives, often for less fit patients, include sequential CRT and RT alone. Many reports describing the relationship between overall survival (OS), toxicity, and dosimetry are based on clinical trials, with strict criteria for patient selection. We performed an institutional analysis to study the relationship between dosimetric parameters, toxicity, and OS in inoperable patients with stage III NSCLC treated with (hybrid) IMRT/VMAT-based techniques in routine clinical practice. Eligible patients had undergone treatment with radical intent using cCRT, sCRT, or RT alone, planned to a total dose ≥ 50 Gy delivered in ≥15 fractions. All analyses were performed for two patient groups, (1) cCRT (*n* = 64) and (2) sCRT/RT (*n* = 65). The toxicity rate differences between the two groups were not significant, and OS was 29 and 17 months, respectively. For sCRT/RT, no dosimetric factors were associated with OS, whereas for cCRT, PTV-volume, esophagus V50 Gy, and contralateral lung V5 Gy were associated. cCRT OS was significantly lower in patients with esophagitis ≥ G2. The overall rate of ≥G3 pneumonitis was low (3%), and the rate of high-grade esophagitis the OS in this real-world patient population was comparable to those reported in clinical trials. Based on this hypothesis-generating data, more aggressive esophageal sparing merits consideration. Institutional auditing and benchmarking of the planning strategy, dosimetry, and outcome have an important role to play in the continuous quality improvement process.

## 1. Introduction

Stage III (locally advanced) non-small cell lung cancer (NSCLC) represents approximately 20% of all new lung cancer diagnoses [1,2]. Many patients are elderly with significant co-morbidities, and multidisciplinary tumor board review is recommended to determine the best treatment strategy [3,4]. In the Netherlands, only a small minority of these patients receive surgery. Surgery is often not feasible due to primary tumor extent, multi-level nodal involvement, or not recommended due to age, comorbidities, or patient’s preference [5,6,7]. In fit patients who do not receive surgery, concurrent chemoradiotherapy (cCRT) is preferred over sequential CRT (sCRT) or RT alone, due to better survival. Palliative treatment, for example, with radiotherapy, or best supportive care may be recommended for patients with severely limited performance status, or excessive co-morbidities. The PACIFIC trial recently reported that combining cCRT with consolidation durvalumab improved 4-year overall survival (OS; 49.6% vs. 36.3% for placebo) [8]. This combination has now replaced cCRT alone as the standard of care. Despite this, the use of cCRT in stage III NSCLC varies significantly, with concerns about treatment toxicity being a contributory factor [8,9,10]. A recent systematic review of randomised trials that compared cCRT with sCRT in patients with unresectable NSCLC reported that while the incidence of grade ≥3 radiation pneumonitis, cardiac events, and treatment-related death did not differ, there was significantly more grade ≥3 acute esophagitis with cCRT (22.2% vs. 6.4%) [11]. However, many of the included studies used 3D conformal radiotherapy. More advanced radiation delivery techniques such as intensity-modulated radiation therapy (IMRT) and volumetric modulated arc therapy (VMAT), can allow for better sparing of important organs at risk (OAR) including the esophagus, lungs, and heart, while hybrid IMRT/VMAT techniques are especially suited to reduce the (contralateral) lung doses [12,13]. Although such IMRT techniques might reduce toxicity and improve outcomes [14,15,16], just using an advanced technique is not enough since clinical toxicity rates can still be high if treatment plans are suboptimal or if sufficient sparing of OAR’s is not possible because of tumor volume or location [17]. Variation in VMAT/IMRT plans has been demonstrated in other tumor sites and can, for instance, be due to choices in OAR sparing and PTV coverage [13,18]. For example, some teams might prioritize esophagus or heart sparing, while others may focus on minimizing mean lung dose or other parameters. Such choices may also affect the resulting toxicity profiles. 

IMRT/VMAT for locally advanced lung cancer was implemented at our center 10 years ago, and treatment planning has consistently focused on minimizing the volume of the contralateral and total lung minus PTV receiving doses ≥5 Gy, and total lung minus PTV receiving ≥20 Gy with the aim to limit high grade radiation pneumonitis [12]. The initial technique was a hybrid-IMRT/VMAT arrangement, switching more recently to full VMAT, still with the emphasis on lung sparing.

Many reports describing the relationship between overall survival, toxicity, and dosimetry in patients with locally advanced NSCLC are based on clinical trials, with strict criteria for patient selection, including good performance status, pulmonary function, etc. [18,19]. These trials have not always mandated the use of IMRT/VMAT [19,20,21]. We therefore performed an institutional analysis to study the relationship between dosimetric parameters and overall survival and toxicity in patients with inoperable stage III NSCLC treated with IMRT/VMAT-based techniques in routine clinical practice. 

## 2. Materials and Methods

Details of patients with stage III NSCLC between 2015 and 2017, prior to the introduction of immunotherapy, were retrospectively retrieved from an institutional database [22]. Patients gave consent to the use of their data and the study was approved by the Institutional Review Board. Patients were eligible for analysis if they were medically or technically inoperable (i.e., surgery was not part of their treatment strategy determined at the tumor board) and had undergone treatment with radical intent using cCRT, sCRT, or RT alone, planned to a total dose ≥ 50 Gy delivered in ≥15 fractions. Patients were excluded if they subsequently underwent surgery or had prior radiotherapy to the thorax or neck. Multidisciplinary tumor board recommendations were consistent with ESMO guidelines [23]. The following IMRT/VMAT approaches were used during this period: (i) Hybrid-IMRT (hIMRT), (ii) full VMAT (RapidArc; Varian Medical Systems, Palo Alto, CA, USA), or (iii) hybrid-VMAT (hVMAT) [12,13]. For the present analysis, toxicity follow up was until May 2020 and survival until October 2020. Esophageal, pulmonary, and cardiac events were retrospectively scored according to the Common Terminology Criteria for Adverse Events (CTCAE v. 5.0) [22]. Tumor staging was done according to the American Joint Committee on Cancer (AJCC) [24].

Additional data on radiation planning parameters were extracted from the departmental radiation oncology information management system (ARIA, Varian Medical Systems). All radiotherapy was planned using a 4-dimensional CT scan. The planning target volume (PTV) was created from the internal target volume (ITV) with the addition of (typically) a 1 cm margin. For cCRT, a dose of 60 or 66 Gy in 30 or 33 fractions of 2 Gy was delivered. Hypofractionated schemes were typically used for sCRT/RT, e.g., 23 or 25 fractions of 2.6 Gy. In general, the planning objectives were to keep the spinal cord dose < 50 Gy, limit the volume of total lung minus PTV receiving at least 20 Gy (V20 Gy, preferably ≤35%), and also to limit both the total and contralateral lung V5 Gy, preferably to <60% and <45%, respectively. When the V20 Gy exceeded 35%, clinicians could decide to accept it, or consider reducing the margin, or the prescribed dose. In the hIMRT and hVMAT techniques, these planning goals were achieved by delivering most of the dose via antero-posterior/postero-anterior fields, with some additional dose from an oblique field; and the IMRT or VMAT component was about 15% of the total dose, primarily to homogenize the PTV [12,13]. With full VMAT delivery, an avoidance sector of 100–120° was used to avoid beams entering through the contralateral lung, in combination with optimization objectives on the contralateral lung and total lung minus PTV. The full VMAT results in similar lung dose distributions to hVMAT, with the main difference being that the esophagus can be spared better. From early 2017, additional planning goals were specified for the full VMAT plans, to try and reduce doses above 35 Gy in the esophagus, and limit the maximum esophageal dose to <100% of the prescribed dose. During this period, no optimization objectives were used for limiting heart dose. All treatments were delivered using a daily image-guided setup on the spine.

The following dosimetric data were collected: Mean PTV dose (Gy), PTV volume (cm^3^), mean lung dose (Gy), total lung minus PTV V20 Gy (%), V10 Gy (%), and V5 Gy (%), contralateral lung V5 Gy (%), esophagus V40 Gy (cm^3^), V50 Gy (cm^3^), V60 Gy (cm^3^), and V65 Gy (cm^3^), mean heart dose (Gy), heart V5 Gy (%), V25 Gy (%), V40 Gy (%), V60 Gy (%), smoking packyears, N-stage, WHO performance score, gender, age, radiotherapy technique (IMRT, hIMRT, hVMAT, fullVMAT), treatment received (cCRT, sCRT, and RT), and total delivered dose (Gy). For the esophagus, absolute volumes were reported because typically, it was contoured only in those slices containing PTV. For lung doses we used lung minus PTV, which is consistent with the international ESTRO ACROP guideline [25]. Most of these variables are consistent with those used in the meta-analysis by Palma et al. [26]. 

All analyses were performed for 2 patient groups, stratified by a priori risk of toxicity [27], into (1) cCRT and (2) sCRT/RT (R Version 3.5.2(2018-12-20)). sCRT and RT were combined because numbers were limited, and the expectation was that these groups would have similar toxicity risk. The Mann–Whitney-U test was performed to identify potential differences between the groups. Univariate logistic regression models were used to investigate associations between dosimetric factors and the development of esophagitis grade ≥ 2 within 90 days. The odds ratio (OR) and corresponding 95% CI were calculated. OS was estimated with Kaplan–Meier curves, calculated from the date of the first radiotherapy treatment to the date of death. OS was compared using the log-rank test for the 2 groups with esophagitis grade ≥ 2 and esophagitis grade ≤ 1. Univariate Cox regression models were used to assess whether dosimetric factors to the esophagus, lungs, and heart affected OS. The hazard ratio (HR) and corresponding 95% confidence interval (CI) for death were calculated. The Kaplan–Meier curve was used to estimate the cumulative incidence of pneumonitis grade ≥ 2 and cardiac events of any grade. The HR or OR was considered statistically significant when the 95% CI was completely above or below 1.0, or when the *p*-value was below 0.05. An HR > 1.0 indicated a worse time to event, an OR > 1.0 indicated higher odds for the event.

## 3. Results

From a total of 197 patients diagnosed with stage III NSCLC at our regional network between 2015 and 2017, 129 fulfilled the study inclusion criteria, and 44 patients were excluded because they received palliative treatment, had previous radiotherapy to the thorax or neck, or received < 50 Gy. The remaining 24 patients were excluded because they received surgery. Median follow-up time was 29 months (4–65 months) for cCRT and 17 (1–58) months for sCRT/RT. Table 1 summarizes the patient and treatment characteristics. The majority of cCRT patients had a total prescribed radiation dose of 66 Gy in 33 fractions (83%), and the remainder 60 Gy in 30 fractions. The majority of sCRT/RT patients had a prescription of 65 Gy in 25 fractions (72%) and the remainder 55–60 Gy. A total of 13 patients (9 sCRT and 4 cCRT) did not complete their radiotherapy course due to worsening clinical condition (7 patients), disease progression (2) or patient refusal (4). The most commonly used technique was hVMAT (68%). Dosimetric data for the two groups did not significantly differ. Almost all patients (93.8%) achieved a total lung minus PTV V20 Gy ≤ 35% (mean 22%), mean total lung minus PTV dose ≤ 18 Gy (mean 13.7 Gy), and contralateral lung V5 Gy ≤ 45% (mean 16%). Figure 1 and Figure 2 illustrate the variation in lung doses in the order of increasing total lung minus PTV V20 Gy, and variation in esophagus doses in the order of increasing esophagus V20 Gy, split into those experiencing no or G1 toxicity, or G2–3 toxicity. Mann–Whitney U-tests between the sCRT and RT alone groups resulted in *p*-values of 0.45–0.83 for OS, esophagitis, radiation pneumonitis, and cardiac events, supporting the decision to combine these two groups.

Toxicity rates were comparable between the two study groups (Table 2). The incidence of radiation pneumonitis was generally low with 25% of patients developing ≥ G2 pneumonitis and 2% developing ≥ G3 in the cCRT group. For the sCRT/RT group, this was 17% and 3%. Of all esophagitis events, 96% occurred within 90 days after treatment, and rates of G3 esophagitis were 17% and 11%, respectively, for the cCRT and sCRT /RT cohorts. Univariate logistic regression analysis for predictors of ≥G2 esophagitis is shown in Table 3. Predictive factors were, in addition to esophagus V40–50 Gy, for cCRT: PTV size and contralateral lung V5 Gy and for sCRT /RT: Mean lung dose, total lung minus PTV V20 Gy, and contralateral lung V5 Gy. Multivariate analysis was not considered desirable/reliable because the groups were too small and/or the frequency of events too low. Cardiac events, consisting of arrhythmia and myocardial infarction, were seen in 5 (8%) and 11 (17%) patients receiving cCRT and sCRT /RT, respectively. Cumulative incidence curves for G2 pneumonitis and cardiac events are shown in Figure 3.

The 30-day, 90-day, and 1-year mortality rates were 0/0%, 0/8%, and 22/40% for cCRT and sCRT/RT, respectively. Median OS for cCRT, sCRT (excluding the time spent receiving chemotherapy), and RT were 29, 17, and 17 months, respectively (Figure 4). The Kaplan–Meier curve showed a significantly higher OS for patients with ≤G1 esophagitis compared to ≥G2 esophagitis for cCRT patients (*p* = 0.006), but not for sCRT/RT alone (*p* = 0.57). Because 96% of all esophagitis events occurred within 90 days after treatment, only patients who had an OS of ≥90 days were included in this analysis. For sCRT /RT, doses to the heart, lung, and esophagus were not associated with OS. However, for cCRT, the PTV volume, esophagus V50 Gy, and contralateral lung V5 Gy were associated with OS, as seen in Table 4. In addition, univariate logistic regression models and Kaplan–Meier curves for progression free survival (PFS) are shown in the Appendix A. 

## 4. Discussion

This retrospective study of 129 stage III NSCLC patients, treated with either cCRT, sCRT, or RT alone, showed that (1) toxicity values for cCRT and sCRT/RT were comparable, (2) rates of severe pneumonitis were low (2 and 3% ≥ G3 in cCRT and sCRT/RT, respectively), (3) esophagitis G2 or 3 had a significant impact on OS in patients undergoing cCRT but not sCRT/ RT, and (4) the parameters related to OS in patients undergoing cCRT were PTV-size, high doses to the esophagus and contralateral lung V5 Gy. 

Compared to similar patients treated between 2003 and 2010 at our institute, the results of the present study show comparable OS values in the sCRT group (17 and 17.4 months). However, in the cCRT group, the OS values were considerably higher than those reported earlier (29 vs. 18.6 months) [28]. The difference might be as a result of several factors, including patient selection; new treatments, such as the use of targeted therapy on progression; more aggressive treatment of (oligo) metastatic disease; standard use of IMRT/VMAT with consistently better lung/OAR sparing [12,13]; or improved dosimetry and daily image guidance. The OS values in our real-world cCRT cohort were comparable to those reported in RCT’s such as in the control arm of the PACIFIC study [8]. We did not observe an improvement of OS in the sCRT/RT group compared to our earlier cohort, possibly because factors such as comorbidity have the largest influence on OS, and not plan quality or OAR doses [7]. These patients may also be less suitable for aggressive therapy on progression. 

A meta-analysis showed an overall rate of symptomatic pneumonitis of 30% and the IMRT-arm of the RTOG0617 trial observed a 3.5% G3 pneumonitis rate, compared to 7.9% for non-IMRT [19,28]. An overview of toxicities reported in the literature can be found in the Appendix A. We postulate therefore that the low rate of pneumonitis that we observed could be due to the use of an IMRT/VMAT technique that consistently focused on sparing the contralateral lung. The comparable toxicity rates between the two groups are of interest since sCRT/RT is generally thought of as being less toxic. However, since they are typically used in patients who are less fit, it is possible that patient factors have raised the toxicity to a level comparable to cCRT. We would have liked to have been able to perform multi-variable analysis to look further at this, and several other outcomes, but the sample size was considered too small for this to be reliable. Considering the lower OS for sCRT /RT, it is important to search for approaches that further reduce the toxicity in this group, possibly by de-intensification of treatment, or with a selective reduction in the dose delivered to the esophagus. Based on the results presented here, and previous planning studies [13], it might be possible to reduce the esophageal dose without having to compromise much on PTV coverage and sparing of other OAR. Not all treatment plans in this cohort attempted to spare the esophagus optimally and therefore it is possible to lower esophageal dose, which might improve OS. 

The esophagitis grade ≥3 incidence of 17 and 11% (cCRT and sCRT/ RT, respectively) in this real-world patient population is similar to that reported in meta-analyses of clinical trial data; see also Appendix A [11,29,30]. The appearance of esophagitis grade ≥ 2 in cCRT was significantly associated with the esophagus V40 Gy, V50 Gy, the contralateral lung V5 Gy, mean lung dose, and PTV size. In our clinical practice, more attention to esophageal sparing was introduced in 2017, with the intention of reducing the chance of high-grade esophagitis. In patients with more contralateral positive lymph nodes, both the esophagus and the contralateral lung will receive a higher dose, which could explain the association with contralateral lung V5 Gy. The lower OS after cCRT for patients with esophagitis grade ≥2 is consistent with a recent study where dose and margins in the involved nodes were reduced, reducing the G3 esophagitis from 12.9% to 3.6% while OS increased from 26 to 35 months [30]. If the esophagus is within the PTV and a minimum PTV dose is specified, more advanced delivery techniques, such as proton beams, may be unlikely to further reduce esophagitis.

No significant dosimetric predictors for pneumonitis grade ≥3 and any cardiac event were found, possibly due to the low incidence of these events, relatively small sample size, and challenges of retrospective toxicity scoring. Previous work reported cardiac doses to be predictors for OS, with V50 Gy heart as the strongest predictor, and cardiac radiation dose exposure has also been identified as a risk factor for major cardiac events [31,32,33]. 

We acknowledge limitations in this retrospective, single-institution study, with modest patient numbers and a lack of multi-variable analysis. In addition, findings in this study might be influenced by our institutional treatment planning approach and might therefore not be readily generalizable to other institutions. As with any dosimetric study, it is possible that dosimetric parameters interact with each other, thereby making it more challenging to identify the key dosimetric predictors of specified outcomes. 

Data from the present study may be used to benchmark the potential of newer techniques such as proton therapy and MRI-guided radiotherapy should they be adopted by our institution. Several planning studies in patients with lung cancer have reported on the dosimetric advantages of proton radiation therapy, with significant dose reduction to OARs such as the lungs, esophagus, and heart [23,26,27,28]. While this is promising, we recently reported, for example, that pre-existent co-morbidity was the key driver of mortality in the first 24 months following diagnosis [7,34,35,36]. This indicates that a combination of patient-derived and dosimetric parameters could more likely be able to predict outcomes than either alone. 

## 5. Conclusions

The use of a hybrid/full IMRT/VMAT technique for high-dose radiotherapy in stage III NSCLC that focused on sparing the contralateral lung resulted in a low (3%) incidence of ≥G3 pneumonitis, and OS and rates of high-grade esophagitis in a real-world patient population, that were comparable with those reported in clinical trials. For cCRT, esophagitis G ≥ 2 was a predictor of OS and was in turn associated with esophageal dose and V5 Gy to the contralateral lung. Full RapidArc is now our standard technique, providing a balance of (contra)lateral lung sparing, esophageal sparing, and smaller dose outside the PTV, especially in the antero-posterior direction where organs such as the heart may be located. Going forward, these hypothesis-generating data suggest that more aggressive esophageal sparing merits serious consideration. We have shown that institutional auditing of the planning strategy, dosimetry, and outcome has an important role to play in the continuous quality improvement process.

## Figures and Tables

**Figure 1 cancers-13-05923-f001:**
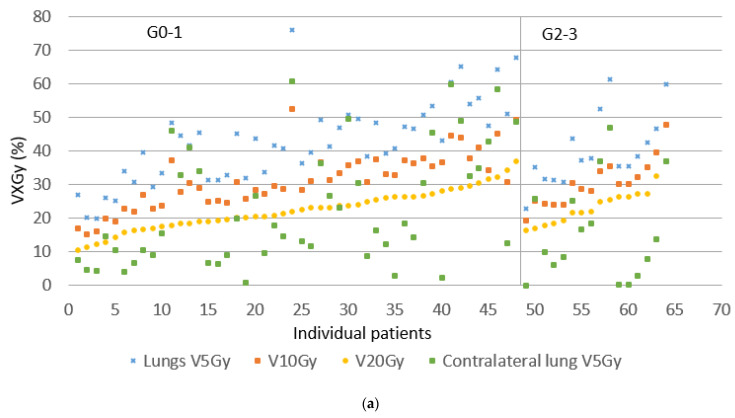
Dosimetric values for cCRT for individual patients. (**a**) Variation in V5–25 Gy total lung minus PTV and V5 Gy contralateral lung, ordered according to an increasing V20 Gy total lung minus PTV, separated in grades 0–1 and 2–3 radiation pneumonitis. (**b**) Variation in V20–60 Gy esophagus, ordered according to an increasing V20 Gy esophagus, separated in grade 0–1 and 2–3 esophagitis.

**Figure 2 cancers-13-05923-f002:**
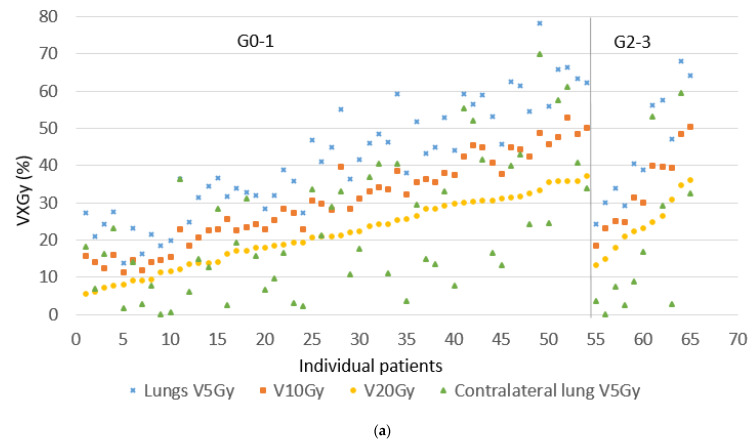
Dosimetric values for sCRT/RT for individual patients. (**a**) Variation in V5–25 Gy total lung minus PTV, and V5 Gy contralateral lung, ordered according to an increasing V20 Gy total lung minus PTV, separated in grade 0–1 and 2–3 radiation pneumonitis. (**b**) Variation in V20–60 Gy esophagus, ordered according to an increasing V20 Gy esophagus, separated in grade 0–1 and 2–3 esophagitis.

**Figure 3 cancers-13-05923-f003:**
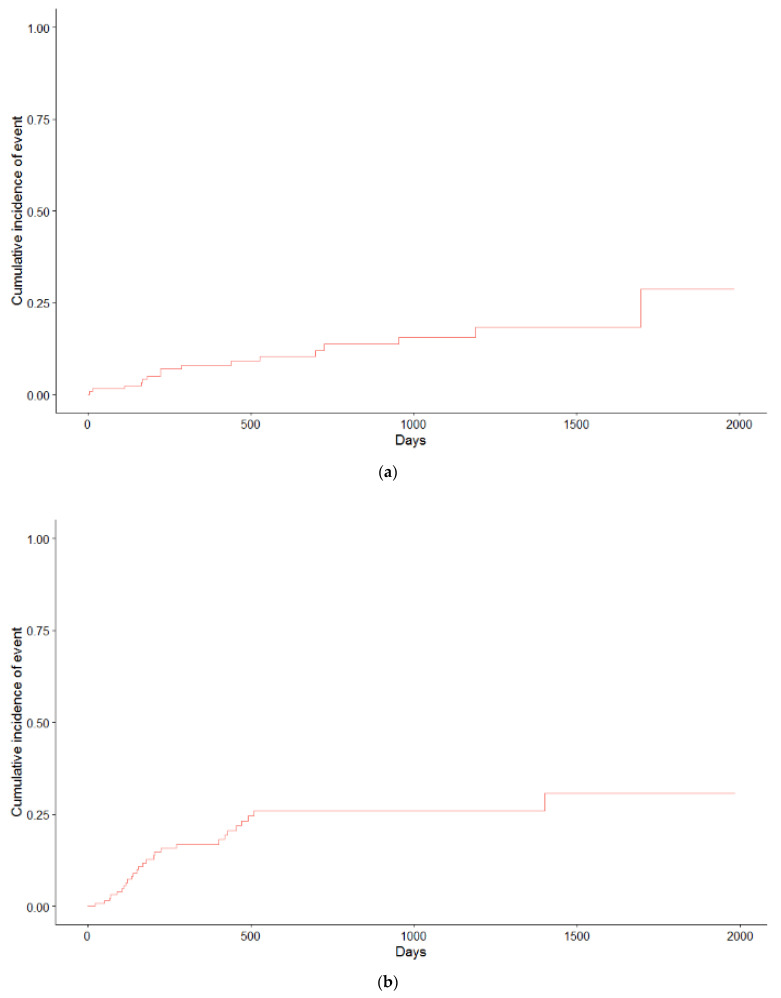
(**a**) Cumulative incidence curve for cardiac events, (**b**) cumulative incidence curve for pneumonitis grade ≥ 2.

**Figure 4 cancers-13-05923-f004:**
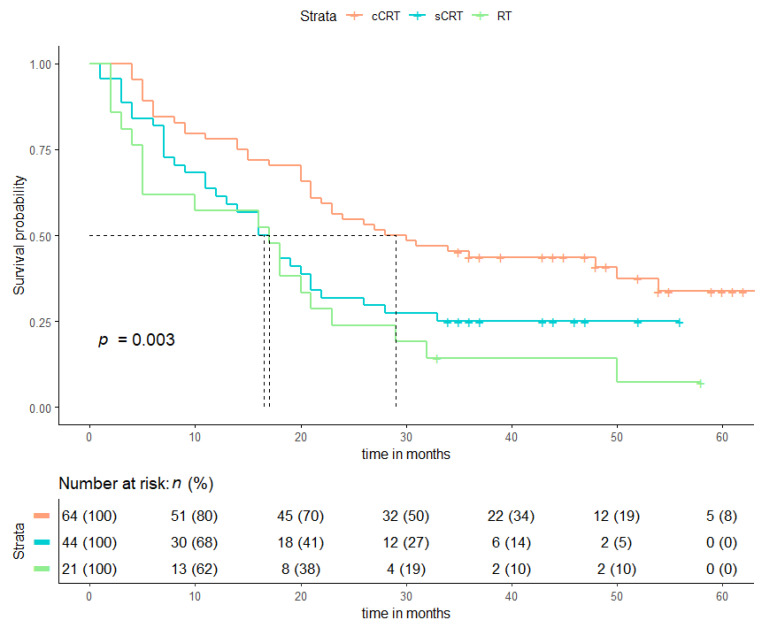
(**a**) Kaplan–Meier curve for Overall Survival (OS) for cCRT and sCRT and RT, (**b**) Kaplan–Meier OS curve for patients with esophagitis grade 0–1 and grade 2–3, for cCRT only, (**c**) for sCRT/RT.

**Table 1 cancers-13-05923-t001:** Patient and radiotherapy characteristics.

Patient Characteristics	cCRT (*n* = 64), *n* (%)	sCRT + RT (*n* = 65), *n* (%)	*p*-Value
Age, median (IQR)	67 (50–89)	67 (33–90)	0.60
Male	32 (50)	36 (55)	0.54
Smoking history	61 (95)	64 (98)	0.31
WHO performance score			<0.01
0	38 (59)	22 (34)	
1	26 (41)	31(48)	
2	-	9 (14)	
3	-	1 (2)	
4	-	-	
Missing	-	2 (3)	
AJCC stage			0.58
IIIA	30 (47)	29 (45)	
IIIB	27 (42)	25 (38)	
IIIC	7 (11)	11 (17)	
T-stage			0.08
T1	16 (25)	7 (11)	
T2	15 (23)	15 (23)	
T3	13 (20)	20 (31)	
T4	19 (30)	19 (29)	
Tx	1 (2)	4 (6)	
N-stage			0.85
N0	5 (8)	6 (9)	
N1	3 (5)	6 ()	
N2	39 (61)	34 (52)	
N3	17 (27)	19 (29)	
Prescribed total dose			<0.01
66 Gy (33 × 2 Gy)	53 (83)	-	
65 Gy (25 × 2.6 Gy)	-	47 (72)	
60 Gy (30 × 2 Gy)	11 (17)	5 (8)	
60 Gy (20 × 3 Gy)	-	3 (5)	
59.8 Gy (23 × 2.6 Gy)	-	8 (12)	
57.5 Gy (23 × 2.5 Gy)	-	1 (2)	
55 Gy (20 × 2.75 Gy)	-	1 (2)	
Radiotherapy technique			0.36
hIMRT	2 (3)	2 (3)	
hVMAT	46 (72)	42 (65)	
FullVMAT	16 (25)	21 (32)	
PTV cm³, median (range)	607 (203–2053)	577 (48–1659)	
Radiotherapy characteristics	Median (range)		
PTV mean dose (Gy)	66.6 (48.4–70.2)	65.3 (10.6–68.7)	<0.01
Mean lung dose (Gy)	13.8 (6.5–22.0)	13.5 (1.2–20.2)	0.60
Mean lung dose ipsilateral (Gy)	30.6 (0.5–48.0)	28.3 (4.2–47.9)	0.27
Mean lung dose contralateral (Gy)	3.5 (0.4–13.8)	3.7 (0.2–16.8)	0.93
V5 Gy lung (%)	41 (10–76)	41 (14–78)	0.93
V10 Gy lung (%)	31 (15–52)	31 (11–53)	0.92
V15 Gy lung (%)	26 (12–42)	26 (9–44)	0.86
V20 Gy lung (%)	22 (11–37)	22 (5–37)	0.74
V25 Gy lung (%)	19 (9–34)	20 (0–32)	0.89
V5 Gy contralateral lung (%)	15 (0–61)	17 (0- 70)	0.73
V20 Gy esophagus (cm^3^)	15.5 (0–51.2)	14.4 (0–48.0)	0.79
V40 Gy esophagus (cm^3^)	11.1 (0–50.0)	10.4 (0–36.1)	0.87
V50 Gy esophagus (cm^3^)	8.4 ( 0–47.7)	7.6 ( 0–30.0)	0.84
V60 Gy esophagus (cm^3^)	4.6 (0–31.2)	3.9 (0–21.2)	0.70
V65 Gy esophagus (cm^3^)	0.6 (0–19.0)	0.1 (0–13.5)	0.07
Mean heart dose (Gy)	11.1 (0.8–42.2)	12.9 (0.5–44.2)	0.62
V40 Gy heart (%)	10 (0–58)	12 (0–60)	0.83

Abreviations: cCRT: Concurrent chemoradiotherapy; sCRT: Sequential chemoradiotherapy; RT: Radiotherapy alone; WHO: world health organization; AJCC: American joint committee on cancer; hIMRT: Hybrid intensitymodulated radiation therapy; hVMAT: Hybrid volumetric modulated arc therapy; full VMAT: Full volumetric modulated arc therapy; PTV: Planning target volume; V5 Gy/V10 Gy/V15 Gy/V20 Gy/V25 Gy lung: Percentage of the lung received at least 5/10/15/20/25 Gy; V5 Gy contralateral lung: Percentage of the contralateral lung received at least 5 Gy; V20 Gy/V40 Gy/V50 Gy/V60 Gy/V65 Gy esophagus: Volume (cm^3^) of the esophagus received at least 20/40/50/60/65 Gy.

**Table 2 cancers-13-05923-t002:** Incidence of pneumonitis, esophagitis, and cardiac events.

Toxicity	cCRT (*n* = 64), *n* (%)	sCRT/RT (*n* = 65), *n*(%)	
Pneumonitis grade			*p*-value
0–1	48 (75%)	54 (83%)	0.18
2	15 (23%)	9 (14%)	
3	0 (0%)	2 (3%)	
4	1 (2%)	0 (0%)	
Esophagitis grade			
0–1	38 (59%)	34 (52%)	0.20
2	15 (23%)	24 (37%)	
3	11 (17%)	7 (11%)	
4	0 (0%)	0 (0%)	
Cardiac event			
No cardiac event	59 (92%)	54 (83%)	
Any cardiac event	5 (8%)	11 (17%)	0.12

**Table 3 cancers-13-05923-t003:** Univariate logistic regression analysis for esophagitis grade ≥ 2. PTV per 100cc increase; Mean lung dose per 100 cGy increase; V20 Gy lung (total lung minus PTV) per 10% increase; V5 Gy contralateral lung per 10% increase.

cCRT				sCRT/RT		
Univariate Variables	OR	95% CI	*p*-Value	OR	95% CI	*p*-Value
WHO performance score ≥ 2	-	-	-	2.61	0.72–10.78	0.15
PTV volume (cc)	1.30	1.07–1.64	<0.01	1.08	0.92–1.28	0.33
Packyears smoking	1.02	0.99–1.05	0.12	0.99	0.98–1.01	0.82
Mean lung dose (cGy)	1.20	1.02–1.44	0.03	1.16	1.04–1.30	<0.01
V20 Gy lung (%)	2.05	0.86–5.27	0.12	2.09	1.16–4.00	0.02
V5 Gy contralateral lung (%)	1.66	1.20–2.40	<0.01	1.36	1.02–1.87	0.05
V20 Gy esophagus (cm^3^)	1.01	1.00–1.02	0.07	1.01	0.99–1.02	0.11
V40 Gy esophagus (cm^3^)	1.01	1.00–1.02	0.02	1.01	1.00–1.03	0.04
V50 Gy esophagus (cm^3^)	1.02	1.00–1.03	<0.01	1.02	1.00–1.03	<0.01
V60 Gy esophagus (cm^3^)	1.02	1.00–1.03	0.06	1.03	1.01–1.04	<0.01
N-stage 2	0.93	0.20–5.10	0.93	1.78	0.47–7.72	0.38
N-stage 3	1.88	0.34–11.66	0.47	2.75	0.63–13.51	0.19
Radiotherapy technique full VMAT	1.105	0.34–3.49	0.86	0.68	0.23–1.95	0.48
Age	0.97	0.92–1.03	0.34	0.98	0.94–1.03	0.51

Abreviations: WHO: World health organization; PTV: Planning target volume; V 20 Gy lung: Percentage of the total lung minus PTV that received at least 20 Gy; V5 Gy contralateral lung: Percentage of the contralateral lung that received at least 5 Gy; V20 Gy/V40 Gy/V50 Gy/V60 Gy esophagus: Volume (cm^3^) of the esophagus received at least 20/40/50/60 Gy.

**Table 4 cancers-13-05923-t004:** Univariate analysis for overall survival. PTV per 100cc increase; mean lung dose per 100 cGy increase; V25 Gy/V40 Gy/V60 Gy heart per 10% increase; V5 Gy contralateral lung per 10% increase.

cCRT				sCRT/RT		
Univariate Variables	HR	95% CI	*p*-Value	HR	95% CI	*p*-Value
V25 Gy heart (%)	1.11	0.90–1.34	0.34	0.94	0.80–1.11	0.45
V40 Gy heart (%)	1.13	0.99–1.42	0.25	0.94	0.76–1.15	0.53
V60 Gy heart (%)	1.29	0.92–1.81	0.14	1.01	0.72–1.38	0.96
Mean heart dose (cGy)	1.00	0.91–1.11	0.27	0.99	0.96–1.00	0.54
V20 Gy lungs (%)	1.14	0.67–1.93	0.62	0.95	0.71–1.28	0.73
V25 Gy lungs (%)	1.11	0.63–1.95	0.72	0.97	0.70–1.33	0.83
V5 Gy contralateral lung (%)	1.21	1.01–1.45	0.04	0.91	0.78–1.06	0.24
Mean lung dose (cGy)	1.00	0.93–1.11	0.60	1.00	0.91–1.11	0.91
V20 Gy esophagus (cm^3^)	1.02	0.99–1.05	0.12	1.01	0.98–1.04	0.57
V40 Gy esophagus (cm^3^)	1.03	1.00–1.05	0.07	1.00	0.98–1.03	0.80
V50 Gy esophagus (cm^3^)	1.04	1.01–1.07	0.02	0.99	0.97–1.03	0.92
V60 Gy esophagus (cm^3^)	1.03	0.99–1.08	0.13	1.01	0.97–1.05	0.61
N-stage 2	0.51	0.17–1.48	0.21	1.93	0.58–6.38	0.28
N-stage 3	0.57	0.18–1.81	0.34	1.34	0.39–4.82	0.62
Technique full VMAT	1.05	0.51–2.15	0.90	0.65	0.35–1.21	0.18
Age	1.03	0.99–1.07	0.16	1.02	0.99–1.04	0.17
Smoking history	1.01	0.99–1.02	0.14	0.99	0.99–1.01	0.54
PTV volume (cc)	1.22	1.11–1.35	<0.01	1.11	0.82–1.22	0.12

Abbreviations: V25 Gy/V40 Gy/V60 Gy heart: Percentage of the heart that received at least 25/40/60 Gy; V20 Gy/V20 Gy lungs: Percentage of the total lung minus PTV that received at least 20/40/50 Gy; V5 Gy contralateral lung: Percentage of the contralateral lung that received at least 5 Gy; V20 Gy/V40 Gy/V50 Gy/V60 Gy esophagus: Volume (cm^3^) of the esophagus received at least 20/40/50/60 Gy; PTV: Planning target volume.

## Data Availability

Patient data cannot be shared publicly. Remaining raw data are available upon request to the corresponding author.

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
