# Peer review of "Relationship between Treatment Plan Dosimetry, Toxicity, and Survival following Intensity-Modulated Radiotherapy, with or without Chemotherapy, for Stage III Inoperable Non-Small Cell Lung Cancer"

_cancers, 2021, doi:10.3390/cancers13235923_

Round 1

Reviewer 1 Report

    1. Why did patients with stage IIIA have a worst OS and patients with IIIC have best OS regardless of treatment?

    Mean OS in days

    cCRT

    sCRT/RT

    IIIA

    897

    543

    IIIB

    949

    633

    IIIC

    1130

    698

    1. I don’t consider it is reasonable to exclude the patients with subsequently surgery. It current study, the authors only include the patients with technically or medically inoperable disease at diagnosis. However, post-CRT surgery indicated that the prior treatment was effective so the disease status could be converted to operable. This possibly explain why the patients with stage IIIA has worst OS as the patients undergoing surgery with better OS were excluded.

    1. I don’t consider that the reply of mixed sCRT/RT is scientific reasonable due to small number for both groups.

    1. What is the version of AJCC staging?

    1. For cCRT no patients had a PS of 2 or higher. Therefore, PS should be a major confounding factor for analysis.

    1. No p-value in Table 1.

    1. “OS is considered as the gold-standard end-point. When it is used, PFS is usually used as a surrogate for OS to try and get a faster result from a study. This is common when testing a new therapy.” It is not true. To investigate the efficacy in stage III disease, PFS is a better end-point than OS. For example, the primary end-point of PACIFIC study is PFS rather than OS. FDA approved durvalumab based on PFS. As I mentioned previously, OS is confounded by several factor such as drive genes, targeted therapy and immunotherapy. If authors would like to use OS as an end-point in current study, they should provide the information of EGFR/ALK/ROS1 mutation status, PDL1 expression and subsequential treatment and analyze them as the prognostic factors.

    1. I don’t agree “To test if factors are independent predictors of esophagitis, a multivariate analysis would be necessary but unfortunately this is not possible due to the small number of events. We added this in the results.” The number of cases is enough to perform multivariant analysis.

Reviewer 2 Report

The small sample size affected the quality of the data analysis. Other than that, the study was unique and warranted further investigations. I have several minor comments.

Table 3: There were two sets of OR - 95% CI - P-value in the table. Was one set for cCRT and the other set for sCRT/RT? If yes, the authors should clearly indicate in the table.

Table 4: Similar to Table 3, Was one set of HR-95% CI-p-value for cCRT and the other set for sCRT/RT?

Line 134-135. It should be changed to “sCRTand RT were combined because the expectation was that these groups would have similar toxicity risk”.

Round 2

Reviewer 1 Report

I have no more comments.

Author Response

This manuscript is a resubmission of an earlier submission. The following is a list of the peer review reports and author responses from that submission.

Round 1

Reviewer 1 Report

In this retrospective clinical study, the authors reported the relationship between dosimetric parameters and overall survival and toxicity in  patients with stage III NSCLC treated with concurrent or sequential chemoradiotherapy. It was found that the use of intensity modulated radiation therapy (IMRT) and volumetric modulated arc therapy (VMAT) resulted in a low (3%) incidence of ≥G3 pneumonitis with OS and rates of high-grade esophagitis being comparable with those reported in clinical trials.

  1. Line 164-165: “The incidence of radiation pneumonitis was generally low with 25/2% of patients developing ≥G2/G3 pneumonitis in the cCRT group and 17/2% in the sCRT/RT group.” It was unclear what 25/2% and 17/2% meant and where those data were presented in Table 2.
  2. Table 2 was a bit confusing. The author may want to add one more row between “Pneumonitis grade” and “0-1” and move “Pneumonitis grade” to the second row. The second column of the first row should be CCRT (N =64), N(%). The third column of the first row should be SCRT (N = 65), N(%). After, the all other redundant “N(%) (n =64)” and “N(%) (n = 65)” can be removed from Table 2.
  3. Table 4 needs a proper title. The title of Table 4 can be similar to that of Table 3 so as to keep things consistent in the manuscript.
  4. Line 36-37. “OS and rates of high-grade esophagitis comparable with those reported in clinical trials.” What reported results were the data of OS and rates of high-grade esophagitis obtained from this study compared with? The authors should provide reference and a brief summary of those results documented in other clinical trials”.

Reviewer 2 Report

This manuscript entitled “Relationship between treatment plan dosimetry, toxicity and survival following intensity-modulated radiotherapy, with or without chemotherapy, for stage III non-small cell lung cancer” by Isabel F Remmerts de Vries et al to investigate the association between treatment plan dosimetry, toxicity and survival for stage III NSCLC patients undergoing RT +/- CT. This is an interesting topic and the findings in current study could be expected. Generally, whole manuscript is quite different to read as lacking logistical writing and proofreading. Major problems existed for study design and analysis so some results are not reliable.

Major comments:

  1. My main concern is that stage III NSCLC has very diverse tumor extension and treatment options. Generally, neoadjuvant CCRT + surgery may be the standard treatment for IIIA and primary CCRT is for IIIB/C. However, the authors performed the study of all stage III NSCLC without specific stage such as IIIB/IIIC or unresectable stage III NSCLC which may results in bias. The authors may include some patients with stage IIIA with better survivals or may enroll the patients with medically inoperable tumors so operation is not possible after CCRT.
  2. The authors excluded the all the surgery but salvage or primary surgery have different role in lung cancer treatment. The authors completed excluded the patients with possible operation so this cohort in current study may be relatively poor than clinical practice.
  3. Why did authors keep sCRT and RT in the same group? I consider these two groups should be different as RT alone usually was performed in medical unfit patients. The authors should explain the reasons or compare the difference for sCRT and RT in an additional table. Otherwise, there is no reason to keep this two together.
  4. Why did the authors include the patients undergoing > 50Gy. In cCRT group, all the patients undergoing > 60Gy which is the optimal dose without surgery. However, for sCRT/RT group, the dose of RT is general lower than cCRT group. Therefore, the survival analysis should sCRT/RT is worse than cCCRT which is expected.
  5. The titles and format of tables should be revised as they are difficult to read.
    1. Table 1, the authors may do statistical analysis to compare the different of all baseline characteristics between two groups. No ECOG was found PS in table 1.
    2. Table 2, the authors should keep CCRT and SCRT+ RT in the table. In addition, the authors should keep cCRT or CCRT consistently through whole manuscript.
  • Table 3, the authors may replace A and B by cCRT and sCRT/RT.
  1. Why didn’t author analyze PS for cCRT group in table 3.
  1. As too many factors influence OS, the authors should analyze PFS which may be the main endpoint for such study.
  2. Univariate analysis for esophagitis in Table 3 including V20-50Gy and other factors. I consider other factors were associated with V20-50Gy so other factors may not meaningful unless authors can prove that the factors other than V20-50Gy is independent factors to predict esophagitis. Similar weakness in the analysis for OS.

Minor:

  1. Stage III, stage 3 should be consistent
  2. Abstract is very difficult to understand and the authors should re-write it.
  3. Line 46. The reason of “The use of cCRT has increased” is strange. PACIFIC trial is not the reason.
  4. Line 123, (R Version 3.5.2(2018-12-20)) ?
  5. Line 164, The incidence of radiation pneumonitis was generally low with 25/2% of patients developing ≥G2/G3 pneumonitis in the cCRT group and 17/2% in the sCRT/RT group. What are “25/2%” and “17/2%” ?
  6. Line 96% is not the formal writing.
  7. Line 217-, The authors discussed possible factors leading to better OS in cCRT group. However, the same factors existed in sCRT/RT group. Why did these factors not influence OS for sCRT/RT group.
  8. Line 236, Based on the results presented here, esophagus dose reduction in cCRT may also be possible without having to compromise on the gain in OS. This paragraph confused me. The authors have found esophagitis can predict better OS and here authors would like to decrease the occurrence of esophagitis without compromise OS. Why and how ?